# Wind-Driven Hydrodynamics in the Shallow, Micro-Tidal Estuary at the Fangar Bay (Ebro Delta, NW Mediterranean Sea)

**Marta F-Pedrera Balsells** *[ID], **Manel Grifoll**[ID], **Manuel Espino, Pablo Cerralbo** and **Agustín Sánchez-Arcilla**

Maritime Engineering Laboratory (LIM), Catalonia University of Technology (UPC), 08034 Barcelona, Spain; manel.grifoll@upc.edu (M.G.); manuel.espino@upc.edu (M.E.); pablocerralbo@gmail.com (P.C.); agustin.arcilla@upc.edu (A.S.-A.)
* Correspondence: marta.balsells@upc.edu

**Abstract:** This article investigates water circulation in small-scale (~10 km$^2$), shallow (less than 4 m) and micro-tidal estuaries. The research characterizes the hydrodynamic wind response in these domains using field data from Fangar Bay (Ebro Delta) jointly with three-dimensional numerical experiments in an idealized domain. During calm periods, field data in Fangar Bay show complex water circulation in the inner part of the estuary owing to its shallow depths and positive estuarine circulation in the mouth. Numerical experiments are conducted to investigate wind-induced water circulation due to laterally varying bathymetry. For intense up-bay wind conditions (wind intensities greater than 9 m·s$^{-1}$), an axially symmetric transverse structure occurs with outflow in the central channel axis and inflow in the lateral shallow areas. These numerical results explain the water circulation observed in Fangar Bay during strong wind episodes, highlighting the role of the bathymetry in a small-scale environment. During these episodes, the water column tends to homogenize rapidly in Fangar Bay, breaking the stratification and disrupting estuarine circulation, consistent with other observations in similar domains.

**Keywords:** Fangar bay; wind; shallow and small-scale bay; estuary; micro tidal

## 1. Introduction

Water circulation induced by local wind effects can contribute to the water exchange between an estuary and the adjacent open sea [1]. Seminal investigations presented by Csanady [2] described how wind stress applied at the surface of a basin of variable depth sets up a circulation pattern characterized by coastal currents in the direction of the wind, with return flow occurring over the deeper regions. Narváez et al. [3] showed that the observed transverse partition of the subtidal circulation is consistent with that driven by local wind in a channel with lateral depth variations. Wong et al. [4] showed how the impact of the wind-induced coastal sea level at the entrance of an inlet exerts a huge impact on the subtidal variability within the interior of the bay. The influence of wind can also have effects on salinity structure or resuspension events in shallow estuaries [5–8]. These studies indicate that spatial scales and water depth determine the flow response to wind forcing. Moreover, Noble et al. [9] determined that both river and wind-driven flow patterns change as a function of water depth in a shallow estuary (below 4 m). In this case, for wind-driven flows the critical parameter is the degree of stratification in the lower bay. However, the particular role that plays the geometry and the bathymetry in wind response in a small-scale bay is still not clear. An illustrative contribution presented by Sanay and Valle-Levinson [10] reflects the role of the bathymetry in the water contribution for larger

and deeper basins than that mentioned in this study. Numerical experiments reveal a wind-induced water circulation dominated by an axially symmetric transverse structure due to laterally varying bathymetry. So, the investigation at very small-scales (order of few km) and shallow depths (few meters) still presents challenges that may be addressed by complementing observational and numerical efforts. The short time scales expected due to the shallowness, the role that plays the geometry (for instance the mouth width) and the link between bathymetry and water circulation are challenging questions to describe properly small-scale bays under micro-tidal conditions.

Contributions focusing on coastal bays and estuaries conclude that they are regions with complex hydrodynamics due to multiple forcing [7,9,11]. In some cases, wind stress competes with baroclinicity to determine the hydrodynamics. However, the astronomic tide [12], seiches [13], water run-off, the bathymetry or the effect of rotation [10] are variables that may not be neglected in the analysis of the local wind influence on this type of water bodies. In these cases, the combination on observational and numerical efforts have allowed obtaining substantial new knowledge on water circulation and termohaline structure.

This paper investigates the relationship between wind and hydrodynamics of small-scale, shallow and micro-tidal estuary. The research combines "in situ" data using the case study of Fangar Bay, located in the Ebro delta, NW Mediterranean Sea, and the implementation of numerical experiments in idealized geometries. These simulations should figure out the impact of the bathymetry on wind-driven water circulation. The complexity of the flow observed have suggested to focus, at this first stage, on the effect of the bathymetry in the wind-driven circulation. Fangar Bay presents problems of water renewal, high water temperatures, anoxia and permanence of pollutants in the water. All these problems affect the oyster and mussel crops present in the bay that need a sustained water renewal for their growth and development [14]. Several intensive field campaigns have been carried out to characterize the bay hydrodynamics as a first step for a subsequent water renewal analysis. In this sense, Fangar Bay is a good example to investigate the hydrodynamics of small-scale and shallow scale bays under micro-tidal conditions.

## 2. Materials and Methods

### 2.1. Study Area

Fangar Bay is part of the Ebro Delta (NW Mediterranean Sea), which extends around 25 kilometres offshore and forms two enclosed bays (Fangar to the north and Alfacs to the south). Both receive freshwater discharges from the irrigation channels of rice fields. The region is micro-tidal and previous analysis in these bays have observed a negligible influence of the tide on the water circulation [5,12]. The main dimensions of Fangar Bay are 12 km$^2$, with a length of about 6 km, a maximum width of 2 km and a volume of water of $16 \times 10^6$ m$^3$ [15]. The average depth is 2 m, with a maximum of 4 m (see bathymetry in Figure 1). The connection with the open sea is located NW and spans approximately 1 km [16]. In recent years, the mouth width has been modified by the accumulation of sediment from the beach located to the north [17], so its width is currently less than 1 km.

The Ebro Delta is influenced by the presence of southerly/south-easterly sea breezes—which do not exceed 6 m·s$^{-1}$ during the summer and spring seasons—and strong winds from the north and west of more than 12 m·s$^{-1}$ in autumn and winter [18,19]. The most frequent wind throughout the year is the NW wind, locally known as the mestral, which is characterized by strong gusts of cold and dry wind [20]. In addition, sporadic easterly winds occur, associated with rainy events [21].

The freshwater contribution is regulated by rice cultivation throughout the year. Open channels occur from April to December receiving a mean flow of 7.23 m$^3$·s$^{-1}$ [22]. This flow is distributed irregularly throughout the year, with maximum values in the months of April to November. Negligible freshwater flow occurs from December to March when the channels are closed [23]. The freshwater flows during these fields campaigns is estimated in 6 m$^3$·s$^{-1}$. This flow is distributed in two main freshwater discharges in Fangar Bay: one in the Illa de Mar harbour and the other, Bassa de les Olles,

located in the bay mouth (Figure 1). Additional freshwater discharges along the coastline are expected because freshwater inputs are regulated by gravity according to sea level. Finally, a substantial contribution from groundwater inputs is expected within the bay [15]. In both cases, the expected freshwater flow is smaller than the water pumping stations mentioned above.

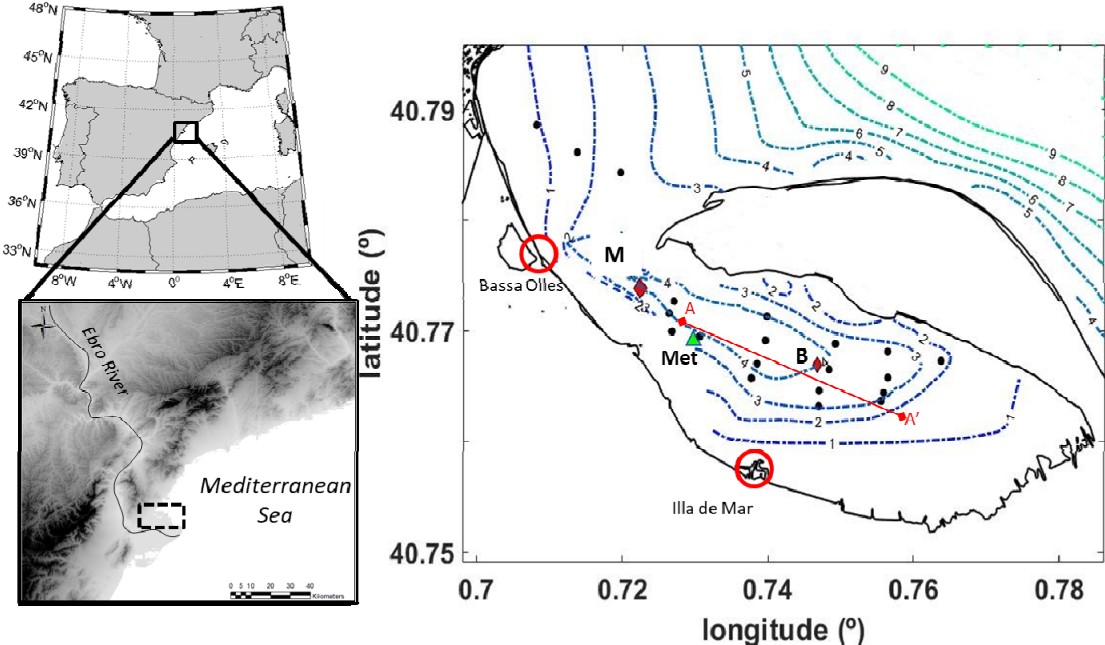

**Figure 1.** Location of the study area. The red circles show the two main points of freshwater discharges (Bassa de les Olles and Illa de Mar), the red diamonds show the points of measurement during the field campaigns (M: mouth; B: bay) and the green triangle shows the location of the meteorological station (Met) during FANGAR-II. The black point shows the conductivity, temperature and depth (CTD) profiles, the blue lines show the bathymetry and the red line (A–A′) shows the longitudinal section presented in Figure 9.

### 2.2. Field Campaigns in Fangar Bay

The observational data corresponded to two field campaigns (each of around two months) from 25 July to 5 September 2017 (FANGAR-I) and from 5 October to 16 November 2017 (FANGAR-II), hence summer and autumn, respectively. The data set consisted of two Acoustic Doppler Current Profilers (ADCPs, from NORTEK, model AQUADOPP 2 MHz) (mooring points M and B in Figure 1), with the velocity and the direction of the water currents obtained every 10 min in 25 cm layers distributed from the bottom to the surface. Moreover, the systems were equipped with pressure systems and a temperature sensor (Vaisala PTB110 and HMP40). A set of CTD (conductivity, temperature and depth, model SeaBird 19plus) campaigns were conducted. During FANGAR-I, 16 CTD campaigns were carried from 11 July to 5 September (two campaigns per week). During FANGAR-II, only two surveys were undertaken, one at the beginning of the campaign (18 October) and the other at the end (16 November). For each of the CTD campaigns, twenty points were chosen, including both the inner and the outside sections of the bay, where temperature and salinity were measured (see black dots in Figure 1).

During FANGAR-I, data from Servei Meteorologic de Catalunya (MeteoCat, http://www.meteo. cat/ (accessed on 30 January 2020)) of the Illa de Buda, located about 12 km from Fangar Bay, were used. This station records wind data every 30 min. During FANGAR-II, a meteorological station (with temperature and humidity sensor, air pressure and acoustic wind sensor) was mounted on a mussel raft near the mouth (Met in Figure 1) to measure wind, atmospheric pressure, air temperature and humidity every 10 min. The measurement periods and instruments are summarized in Table 1.

**Table 1.** Data acquisition instruments and observational periods (year 2017) shown in Figure 1.

| Name (ID) | Observations | Period | Data Interval (min) |
|---|---|---|---|
| Meteo station (Met) (wind Sonic/Vaisala HMP40/Vaisala PTB110) | Wind, atmospheric pressure | 19 October–16 November | 10 |
| Illa de Buda station (wind Sonic/Vaisala HMP40/Vaisala PTB110) | Wind, atmospheric pressure | 25 July–5 September | 30 |
| ADCP mouth (M) (Nortek Aquadopp 2 MHz) | Currents, sea level, waves, bottom temperature | 25 July–5 September / 5 October–16 November | 10 |
| ADCP inner bay (B) (Nortek Aquadopp 2 MHz) | Currents, sea level, waves, bottom temperature | 25 July–5 September / 5 October–16 November | 10 |
| CTD (SeaBird 19plus) | Temperature, salinity | 11 July–5 September / 18 October and 16 November | - / - |

### 2.3. Numerical Modeling

A series of numerical experiments were conducted using the Regional Ocean Model System (ROMS) to analyse the hydrodynamic wind response in small and shallow estuaries. The ROMS numerical model is a 3D, free-surface, terrain-following numerical model that solves the Reynolds-averaged Navier–Stokes equations using hydrostatic and Boussinesq assumptions [24]. ROMS uses the Arakawa-C differentiation scheme to discretize the horizontal grid in curvilinear orthogonal coordinates and finite difference approximations on vertical stretched coordinates [25]. The numerical details of ROMS are described extensively in [24]. This model has been used and validated in similar bays and estuaries, such as Alfacs Bay located south of the Ebro Delta (e.g., [13,26,27]). The domain used is inspired in Fangar Bay and consists in an idealized domain due to the difficulty of achieving good bathymetry. The domain consists of a regular 37 × 27 grid with a horizontal resolution of about 70 m (Figure 2a) and 10 sigma levels in the vertical direction (the bottom layer being the first and the surface layer the tenth, Figure 2b). The model boundary is located 10 points away of the mouth entrance to avoid boundary noise (faster than show in the Figure 2a). The 2D variables were accommodated with the Chapman and Flather algorithms [28], whereas a clamped boundary condition is imposed on the 3D variables. The same model configuration has been used and validated in the bay located south of the Delta, the Alfacs Bay [27].

Figure 2a presents the different geometries used for the numerical experiments. We modeled a simple case, regarding the water body as a channel 2.5 km long and 1.9 km wide, to simplify the response of the wind to the idealized bathymetry similar to that of the Fangar Bay. The first was of a homogeneous depth of 4 m (GEO 1). The second was variable bathymetry (v-shape channel with 1 m to the sides up to 4 m in the central part, GEO 2). The same experiment was then conducted, but reducing the mouth of the channel to simulate the narrow entrance of the Fangar Bay (GEO 3). Finally, it was simulated to include variable bathymetry (GEO 4). Experiments were conducted up-bay wind and down-bay wind according to the axis of the estuary (wind spatially homogenous). An additional simulation took into account the Coriolis effect for GEO 1 to verify the influence of the Earth's rotation. All simulations are summarized in Table 2. Constant temperature (20 °C) and salinity (36) conditions were applied. Freshwater inputs were deactivated in order to observe the behaviour of the currents only as a function of wind and bathymetry. The bottom boundary layer was parameterized with a logarithmic profile using a characteristic bottom roughness height of 0.002 m. The turbulence closure scheme for the vertical mixing was the generic length scale (GLS) tuned to behave as a k-ε [29]. Horizontal harmonic mixing of momentum was defined with constant values of 5 $m^2 \cdot s^{-1}$. Long-term simulations (around 2 months each) were performed in order to analyze the wind response avoiding spurious velocities. The wind is constant throughout the simulation. According to

the results less 24 h of the spin-up is observed in the numerical results. The numerical results used are 16 days after the start of the simulation to avoid spurious velocities.

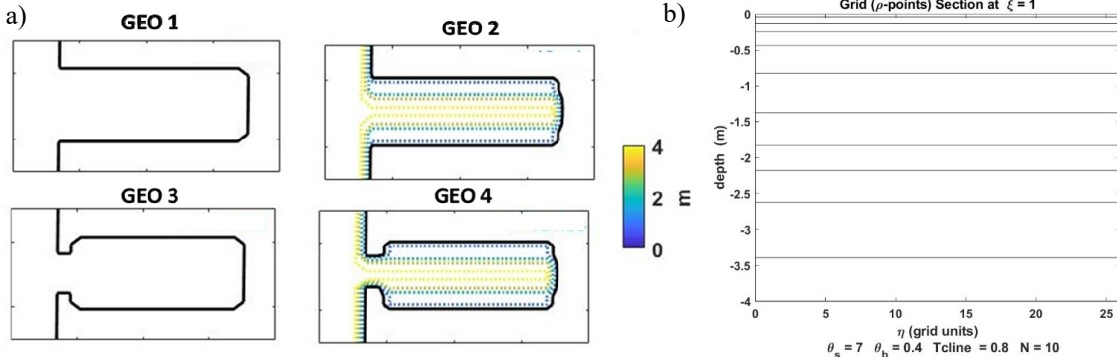

**Figure 2.** (**a**) Geometry and bathymetry for the four configurations used in the numerical experiments. GEO 1 and GEO 3 have 4 m of water depth. (**b**) Sigma layers used in the numerical model. The geometrical parameters are included following the stretching formulation described in Cerralbo et al. (2015).

**Table 2.** Summary of the different simulations used in Regional Ocean Model System (ROMS).

| Simulation Name | Geometry | Wind Direction | Intensity Wind (m·s$^{-1}$) | Coriolis |
|:---:|:---:|:---:|:---:|:---:|
| SIMU 1 | GEO 1 | Up-bay wind | 12 | No |
| SIMU 2 | GEO 2 | Up-bay wind | 12 | No |
| SIMU 3 | GEO 2 | Up-bay wind | 6 | No |
| SIMU 4 | GEO 2 | Down-bay wind | 12 | No |
| SIMU 5 | GEO 3 | Up-bay wind | 12 | No |
| SIMU 6 | GEO 4 | Up-bay wind | 12 | No |
| SIMU 7 | GEO 4 | Down-bay wind | 12 | No |
| SIMU 8 | GEO 1 | Up-bay wind | 12 | Yes |

## 3. Results

### 3.1. Meteorological Description of Fangar Bay

Wind roses for both field campaigns are shown in Figure 3. During FANGAR-I the predominant winds were southerly and easterly, associated with sea breezes, whereas during FANGAR-II the predominant and strongest winds were from the NW (i.e., an upward wind direction considering the axis of the estuary). Note the presence of NW winds in both campaigns.

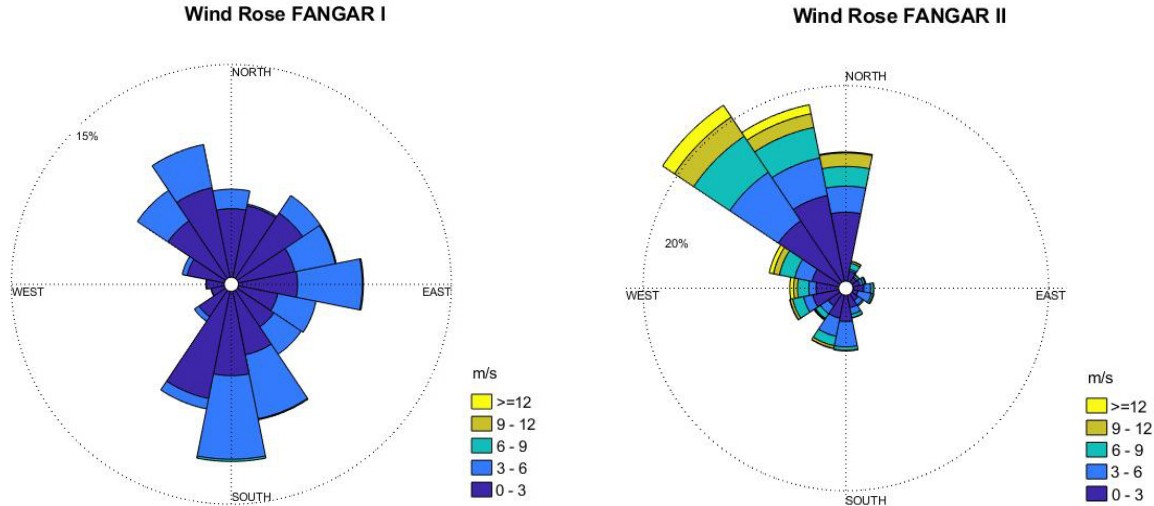

**Figure 3.** Wind roses of mean velocity for FANGAR-I (**left**) and FANGAR-II (**right**) during both campaigns. Wind intensities are grouped by intervals of 3 m·s$^{-1}$.

Figure 4 shows the time series of the wind and air temperature measured at the corresponding meteorological station. During summer (FANGAR-I), the sea breeze was characterized by daily southerlies with a wind intensity of 6 m·s$^{-1}$. Furthermore, two episodes of NW winds were distinguished during FANGAR-I—from 6 to 12 August (henceforth called episode E1) and from 29 August to 3 September (henceforth called E2)—as well as in FANGAR-II from 21 to 23 October (henceforth called E3) and from 5 November until the end of the campaign (henceforth called E4). These NW episodes could also be identified alongside the air temperature drops in Figure 4 superimposed at daily variability during both field campaigns.

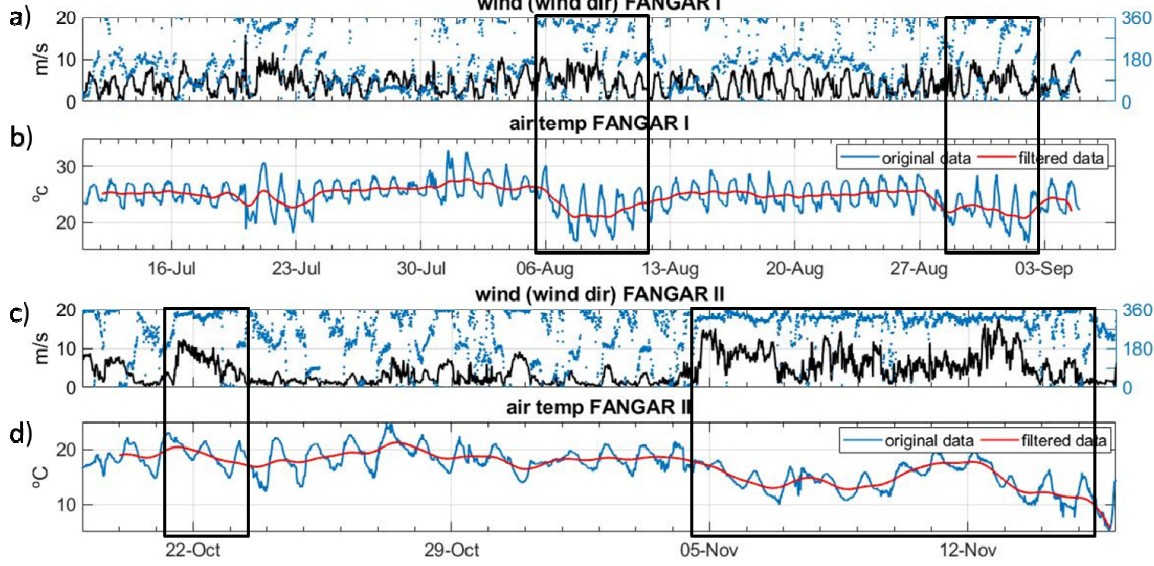

**Figure 4.** Wind direction and intensity from FANGAR-I (**a**) and FANGAR-II (**c**). Air temperature for FANGAR-I (**b**) and FANGAR-II (**d**). Air temperature is filtered to exclude daily variability. The black box shows the periods with falling air temperatures during NW winds.

### 3.2. Hydrodynamic Description of Fangar Bay

A dispersion diagram of the water currents (surface and bottom) measured by the ADCPs in each campaign is shown in Figure 5. The diagram shows how in the mouth (M station) the flow was

aligned following the longitudinal axis of the bay. By contrast, in the B position the flow did not show a prevalent pattern, the water circulation being more complex. In addition, Figure 5 demonstrates how the water flow in the mouth was larger than in the inner bay. The maximum water currents measured in the mouth were 0.5 m·s⁻¹ in FANGAR-I and FANGAR-II, but below 0.02 m·s⁻¹ in both campaigns within the bay. Furthermore, the surface and the bottom signal showed similar patterns across the campaigns.

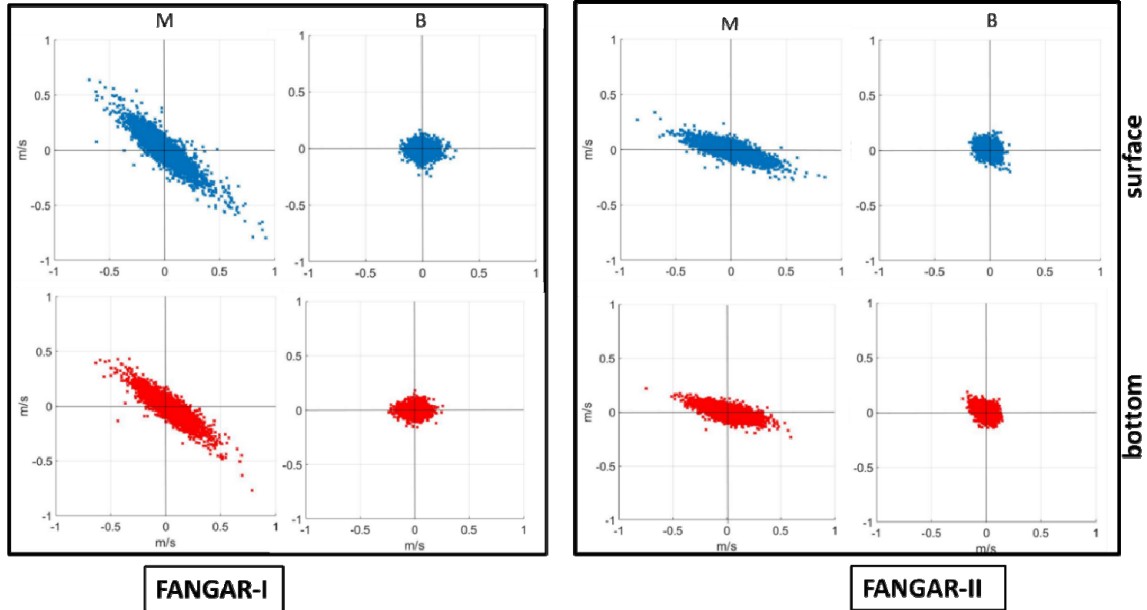

**Figure 5.** Dispersion diagram for FANGAR-I (**left**) and FANGAR-II (**right**), in blue the surface water circulation and in red the bottom water circulation. The first column of panel corresponds to the measurements in the mouth (M) and the second column to those within the bay (B).

Regarding the flow alignment shown in the dispersion diagram (see Figure 5), the water current observations were rotated following the alongshore (longitudinal axis of the bay with positive inward) and cross-shore alignment. The rotation angle was equal to 35° for FANGAR-I and 15° for FANGAR-II (north clockwise negative) according to the flow alignment of the dispersion diagram. Discrepancies in the rotation angles probably owed to differences in the mooring location of the ADCPs. Figure 6 shows the alongshore time-averaged currents for each measurement layer of the ADCPs. The resultant profiles in the water column revealed a predominant two-layer structure during FANGAR-I (Figure 6a), indicating that estuarine circulation was the surface current leaving the bay (negative values), while bottom currents were entering (positive values). During FANGAR-II, the water circulation pattern was similar to FANGAR-I, aside from differences in surface values (above 1 m depth, Figure 6b) when averaging the whole period. However, after differentiating calm wind or sea breeze periods with those of the NW winds (i.e., E3 and E4), distinctions became evident. Indeed, during such periods the profile corresponded to the positive estuarine circulation pattern, normal behaviour in this type of geometry (Figure 6c), whereas when considering only the NW wind episodes that lasted the entirety of November, the water currents were negative (leaving the bay, Figure 6d) in almost all water column broking down the positive estuarine circulation.

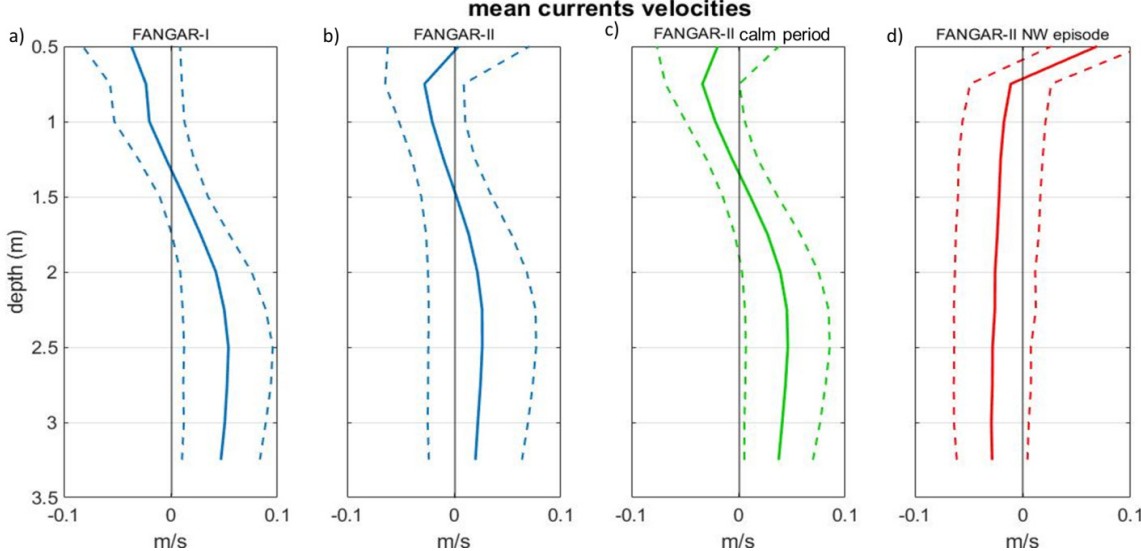

**Figure 6.** Average alongshore current velocity (positive outward) profile (continuous line) and standard deviation (dashed lines) for the mouth in FANGAR-I (**a**) and in FANGAR-II (**b**); (**c**,**d**) show the mean profile during FANGAR-II for the calm period (**c**) and the NW wind period (**d**).

Figure 7 shows the spectral analysis of the alongshore current in the mouth and in the inner bay. High-frequency peaks corresponding to the tidal ranges and typical resonance periods are shown. The seiching band using the Merian formula $(2L/n\sqrt{gH})$ approximately corresponded to periods of two hours in the fundamental mode (considering a characteristic length of about 6 km). These values are consistent with other locations in semi-closed bays [12,29,30].

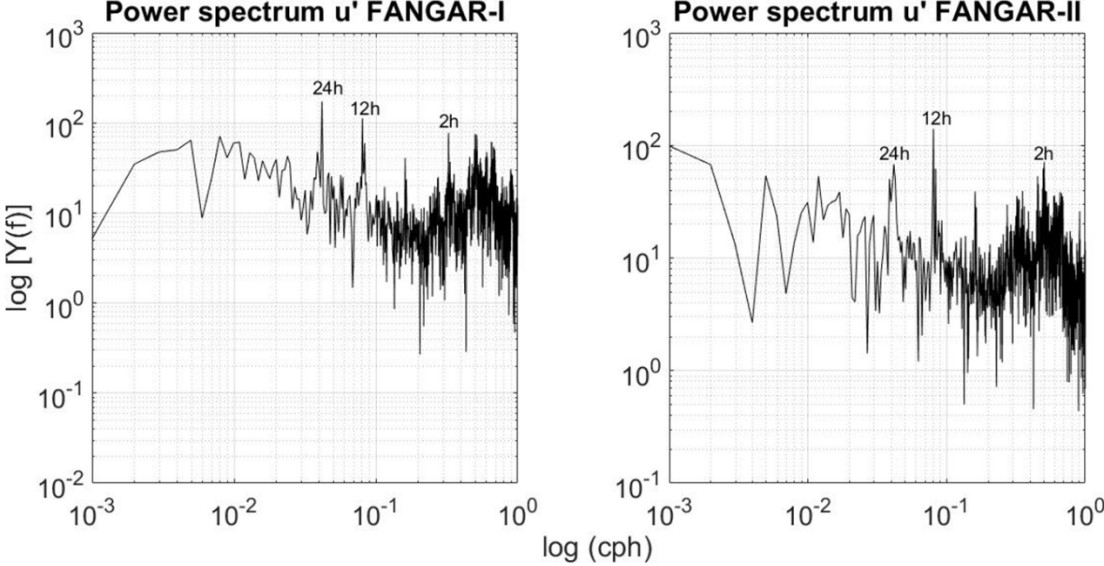

**Figure 7.** Spectral analysis of the signal of the alongshore current in the mouth for FANGAR-I (**left**) and FANGAR-II (**right**).

In order to conduct an episodic analysis, the time series were filtered using a 30 h Lanczos [31] to focus on the low-frequency signal. Figures 8 and 9 show the rotated and filtered components of water currents jointly with wind, temperature and mean flow. Additional measurements of T and S are also shown in Figure 8, with Figures 8d,e and 9c–e presenting the estuarine circulation mentioned previously, whereby the surface water flows out from the bay (negative and blue color) while the bottom water circulation enters (positive and red color). These periods were characterized by a strong stratification. This was due to the fact that the irrigation channels during FANGAR-I were open, contributing to the discharge of fresh water into the bay (Figure 8c). The same occurred at the beginning of FANGAR-II. Marked with a black box are the episodes of NW winds in Figure 8. Note that during these episodes, both temperature and salinity were homogenized (Figure 8b,c) and there was a change from positive estuarine circulation to homogeneous currents in the water column. Moreover, a substantial drop in sea bottom temperature occurred (Figure 8f), notably during episodes E1 and E2 in Figure 8, and during events E3 and E4 in Figure 9. Note that during NW wind episodes, the currents were opposite to the wind direction in both locations (i.e., M and B). Figures 8g and 9g shows the average flow through the mouth. It can be seen that during the NW episodes the net flow is outflow in the central area of the channel, while in the rest of the period occurs the opposite.

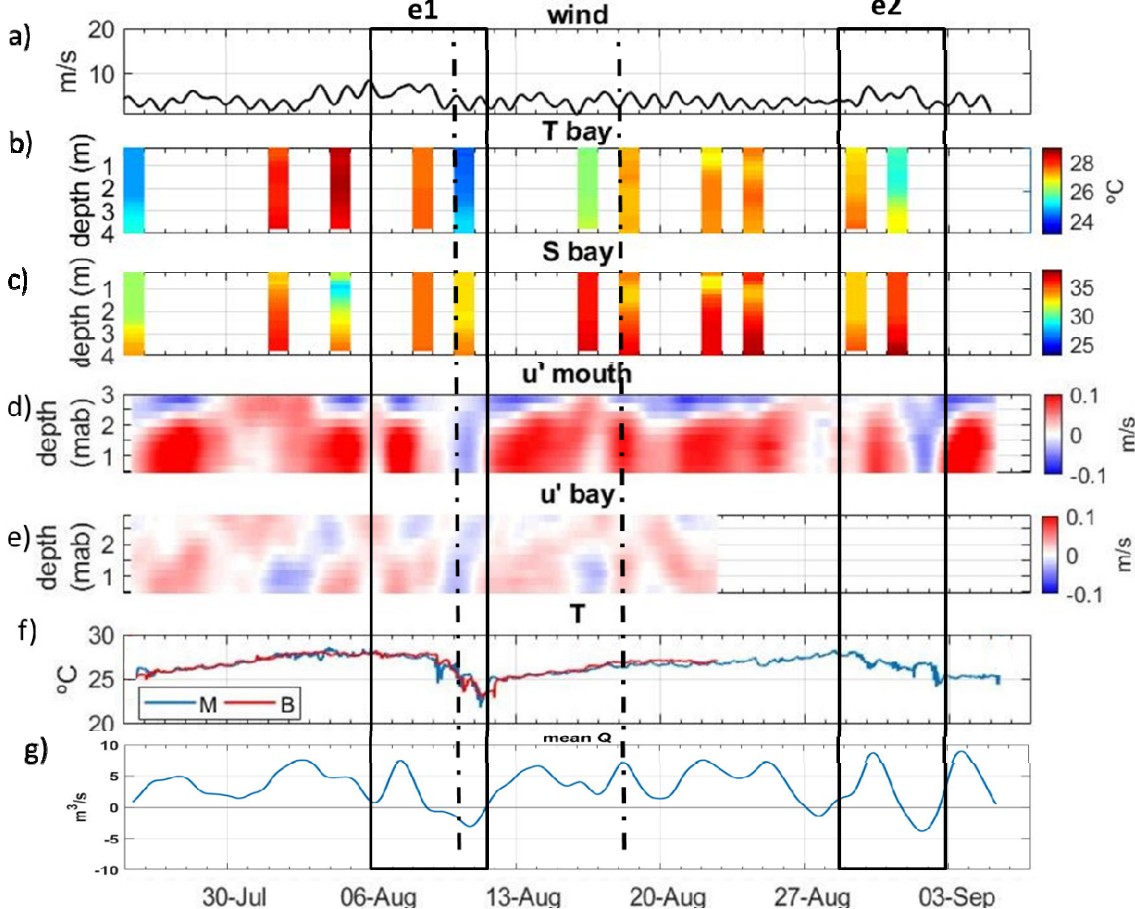

**Figure 8.** From top to bottom, wind speed (**a**), time series of temperature at a central point of the bay (**b**), time series of salinity at a central point of the bay (**c**), alongshore currents in mouth (**d**), alongshore currents in the bay (**e**), and bottom temperature (**f**) in mouth (blue) and bay (red), mean flow (positive inflow) in the mouth (**g**). The black boxes show the NW events. The dotted lines mark the days of the vertical sections shown in Figure 10.

The water temperature observations during FANGAR-I exhibited variations of between 23 °C and 29 °C (Figure 8b and f). The maximum temperature was reached on 4 August after the ambient temperatures reached 30 °C (Figure 8). However, just two days later on 6 August, the water temperature dropped drastically to 23 °C, indicating a difference of 6 °C. This event coincided with the presence of the NW winds, as can be seen in Figure 8. Once the wind stopped, the temperatures rapidly increased again, reaching 28 °C, as can be seen in Figure 8b–f. As noted in the previous paragraph, during episodes E1 and E2, the wind events started on 6 and 27 August, respectively. For both episodes, the temperatures began to drop from those dates and persisted during the subsequent days when the wind died down (Figure 8b–f). This temperature decrease was associated with a cold NW wind, alongside the likely contribution of an open sea entrance (i.e., colder) through the mouth, indicated by the reduction in water temperature observed a few days later, when the wind stopped.

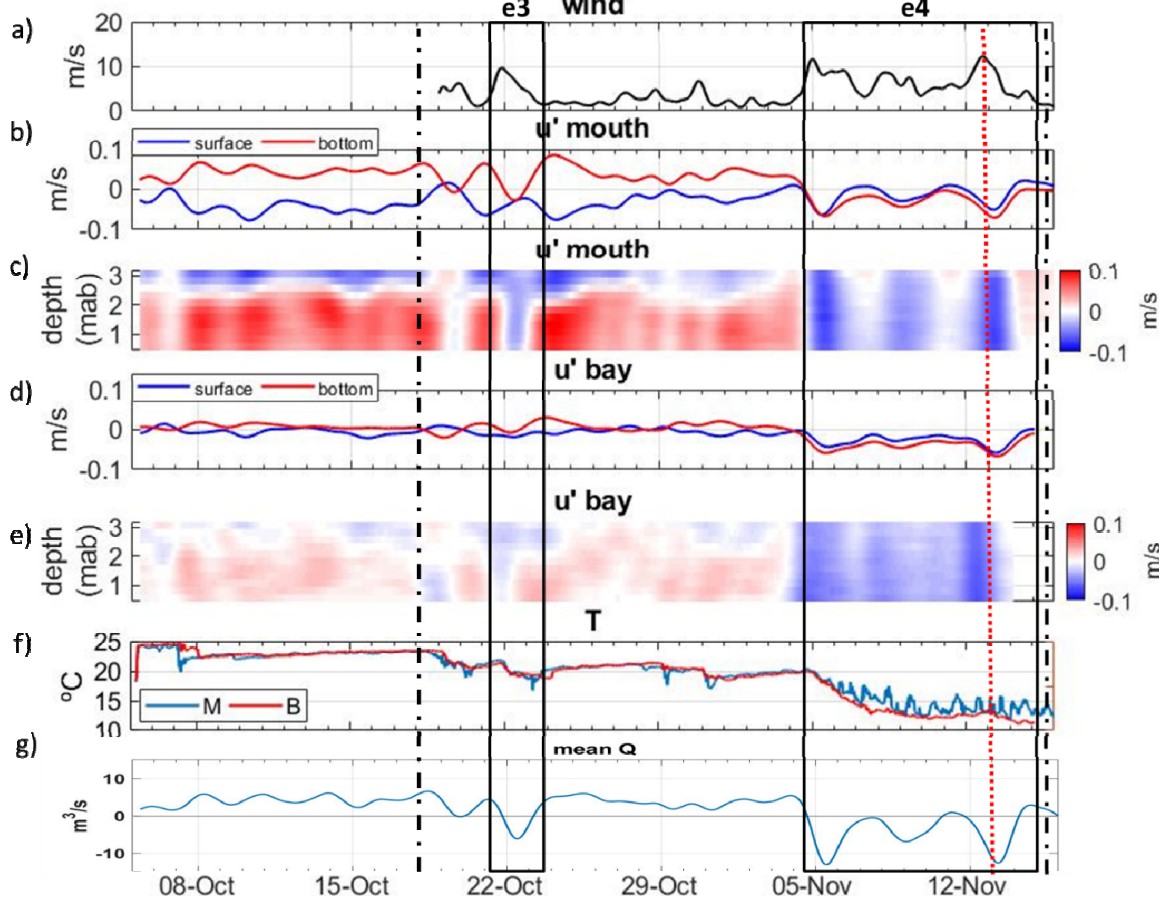

**Figure 9.** From top to bottom, wind speed (**a**), alongshore currents in the mouth (**b**), Hovmöller diagram in the mouth for the alongshore currents (**c**), alongshore currents in the bay (**d**) and Hovmöller diagram in the bay for the alongshore currents (**e**), and bottom temperature (**f**) in the mouth (blue) and the bay (red), mean flow (positive inflow) in the mouth (**g**). The black boxes show the NW events. The dotted lines mark the days of the vertical sections shown in Figure 10.

During FANGAR-II, the water temperature was lower than FANGAR-I because the ambient temperature decreased (i.e., autumn season) (Figure 9a). Throughout FANGAR-II, water temperatures between 11 °C and 25 °C were recorded. The campaign began with water temperatures of 22 °C, until 19 October saw a small decline in the sea bottom water temperature (20 °C) (Figure 9f). This could be associated with E3, when another drop in the water temperature occurred, falling to 18 °C. Once the NW wind stopped, the water temperatures rose slightly again to 20 °C, until 5 November when there was another episode of NW winds (E4). Note that before this event the water temperatures in the bay and in the mouth were similar, whereas from 5 November there were oscillations of about 4 °C in the data from the mouth, as can be seen in Figure 9f. This is due the diurnal cycle of heat fluxes (day/night) during those days.

Figure 10 shows a vertical section of the bay area (A–A′, Figure 1) for four different situations. Figure 10a,b depicts a situation prior to NW winds during FANGAR-I and during FANGAR-II, respectively. In these cases, a low vertical gradient in temperature could be observed (0.5 °C). Focusing on salinity, there was an evident gradient, with less salty surface waters (freshwater contributions, 31–33 for Figure 10a, 32–34 for Figure 10b) and deeper, saltier waters characteristic of ocean waters (37 for Figure 10a, 38 for Figure 10b). Figure 10c,d shows the T/S values during or immediately after NW episodes (i.e., E1 and E4). There was a clear horizontal gradient for both temperature and salinity in Figure 10c (i.e., a decreasing gradient as it enters the inner part of the bay), while in Figure 10d the salinity was homogenized in the water column and the temperature also decreased towards the inner part of the bay.

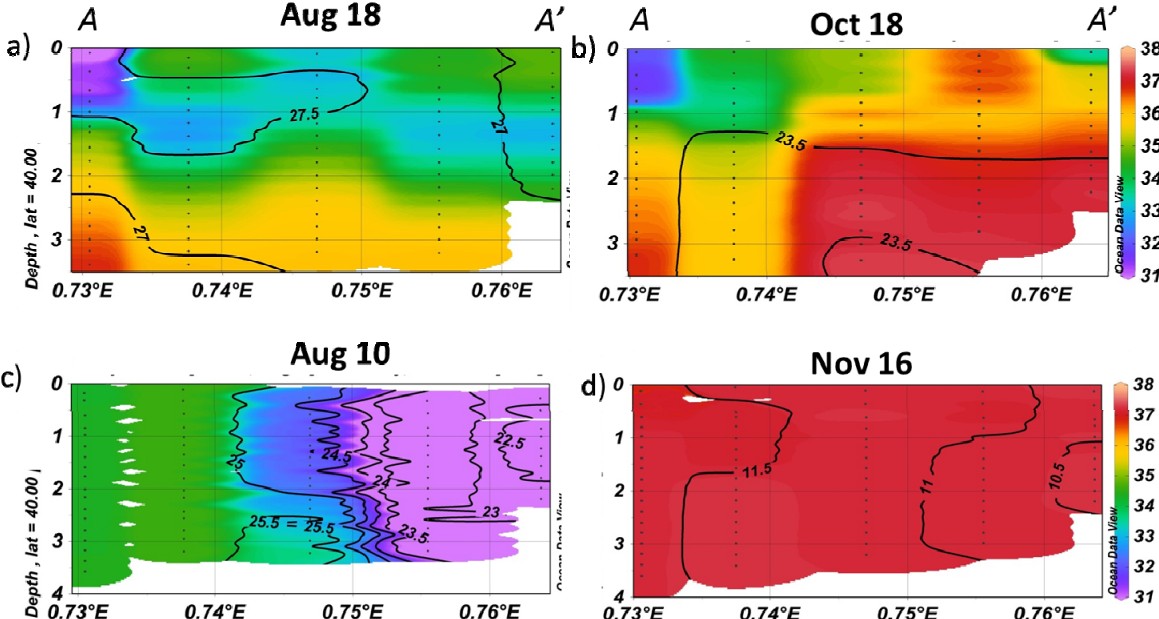

**Figure 10.** Vertical transect of (**a**) stratification summer situation, (**b**) stratification autumn situation, (**c**) situation during episode E1, (**d**) situation during episode E4. The color bar shows salinity values; black lines show temperature values.

### 3.3. Numerical Experiments

Figure 11 shows the results of the numerical model in terms of surface and bottom water currents, with no more variables than the force of the wind pushing the water. SIMU 1 (up-bay wind) indicated an expected water circulation pattern for the homogeneous flow (surface water following the wind direction and return flow at the bottom). However, when a varying bathymetry was considered (SIMU 2), the surface flow in the central axis reduced significantly in comparison to the shallower shore where the bottom friction is very relevant due to its shallowness. Tests have been made with different bottom roughness and it was found that the magnitude varies but the circulation pattern does not. In this case, an outward flow was observed in the central axis of the estuary in the bottom layers, opposite to the direction of the wind, as noted in the observations of the campaigns in the previous section. A homogeneous upward flow is observed at the shore. This pattern was reproduced for upward and smaller winds (SIMU 3). For the case down-bay wind, the picture differed significantly: downward flow towards the shore and upward flow in the axis and the bottom layers (SIMU 4). Therefore, the numerical experiments revealed transversal variability in flow direction due to the bathymetry. SIMU 5, 6 and 7 reproduced the previous cases but a stretching at the bay entrance is added. In this case, the pictures do not significantly differ in relation of the upward/downward flow in the shore/central axis for bottom layers in relation to the open mouth. The Coriolis effect was negligible in idealized cases when comparing SIMU 1 and SIMU 8. Finally, sensitivity test of the bottom friction parametrization has not shown relevant influence on the water circulation patterns affecting mainly the water speed.

Focussing on the impact of the wind on water circulation in the bay entrance, the transversal barotropic velocities are shown in Figure 12. This figure shows how the laterally variable bathymetry induces axially symmetric transverse structure (see inshore velocities near the boundaries and offshore in the central section of the bay entrance for SIMU 2; note that barotropic velocities for SIMU 1 its near to zero). Opposite water circulation pattern occurs for downwind simulation (SIMU 4). Bay mouth stretching (SIMU 6) has a similar pattern than SIMU 4 but increasing the spatial gradients between inflow and outflow (not shown). In this case, the magnitude of the transversal barotropic velocity is lower, reducing also the flow exchange expected between the bay and the open sea if the width mouth is reduced.

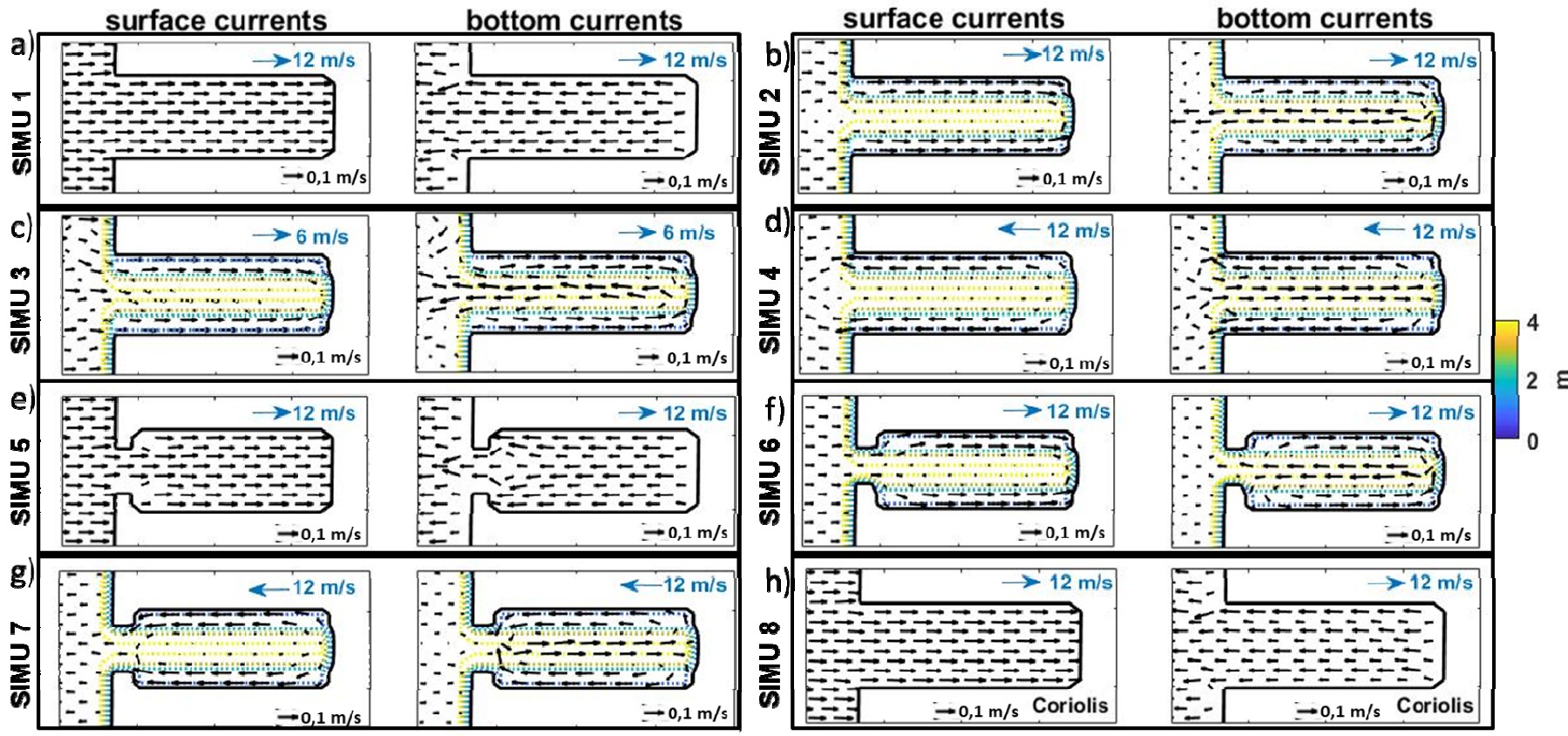

**Figure 11.** Representation of the simulations: (**a**) SIMU 1, surface and bottom; (**b**) SIMU 2, surface and bottom; (**c**) SIMU 3, surface and bottom; (**d**) SIMU 4, surface and bottom; (**e**) SIMU 5, surface and bottom; (**f**) SIMU 6, surface and bottom; (**g**) SIMU 7, surface and bottom; (**h**) SIMU 8, surface and bottom. Blue arrows show the wind direction with the intensity wind and black arrows show the water current intensity.

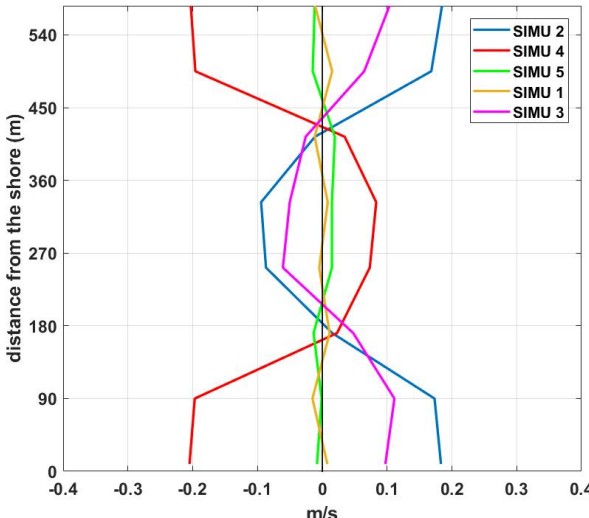

**Figure 12.** Barotropic velocities (negative outflow) in the entrance section for selected simulations.

## 4. Discussion

Local winds have the predominant influence on water circulation in small, shallow and micro-tidal estuaries and coastal bays. Such influence may lead to complex water circulation patterns due to bathymetric and geometric effects. For instance, Geyer [5] has found that geometric constrictions restrict wind-induced circulation, resulting in strong fronts between well-mixed reaches in two small Cape Cod (USA) estuaries. Scully et al. [32] proposed appeared during NW strong wind periods: up-estuary wind weakens and even reverses the shear and reduces the stratification. Li et al. [33] suggested that wind mixing may dominate over wind straining under typical wind forcing conditions in Chesapeake Bay. More recently, Xie et al. [34] tried to explain the apparent contradiction between these two studies and revealed opposite responses of the upper and lower bay regions to axial winds. Furthermore, Coogan et al. [35] have identified substantial transverse variability in Mobile Bay (USA), including the influence of the bathymetric effects previously postulated by [9]. Shallow depths accentuate the wind's influence, sometimes leading to complex water circulation according to observational investigations where the considerable variability of time series is observed [5,7,34]. Llebot et al. [6] have suggested that the origin of high variability of water currents corresponds to wind variability which is more effective in shallow depths in comparison to deep waters. For instance, the sea breeze represents a sequence covering a wide range of wind directions and intensities. Fangar Bay provides a good example of such estuaries (i.e. shallow and micro-tidal), where the bathymetry and the geometry are relevant, as well as being a small-scale bay, which accentuates the action of the wind. The data analysis conducted here, based on two intensive field campaigns, revealed a differentiated pattern in terms of the function of the location (mouth or inner part of the estuary) and the wind conditions (calm or intense). During calm conditions, estuarine circulation linked with strongly stratified conditions were identified in the mouth, associated with strongly stratified conditions. In the inner part, stratification in the water column also occurred, but water circulation did not show a clear pattern. Mixing in the water column occurs during intense wind with substantial modifications in temperature and salinity profiles as will be analyzed later. Particularly during up-bay wind episodes, outward flow conditions were observed in the mouth.

The numerical experiments revealed an axially symmetric transverse structure due to laterally varying bathymetry: up-bay flow occurs in the lateral shoals and down-bay flow in the central channel for up-estuary wind pulses. Consequently, the numerical model's results supported the observed divergence between wind and water flow direction in the center of the mouth in Fangar Bay. This combination of observations and a numerical model confirm that there is a strong transverse

variability on water flow when bathymetric effects are considered. Comparing the SIMU 1 and SIMU 2 numerical experiments provides an illustrative example of the influence of bathymetry (see Figure 11). These results are consistent with Alekseenko et al. [36] for a coastal lagoon in the Mediterranean Sea (opposite flow in comparison to wind direction in a central part of the lagoon). Moreover, Sanay and Valle-Levinson [10] and Narváez et al. [3] found similar results for large bays than the one cited in this paper, highlighting the value of numerical experiments seeking the fundamental role played by bathymetry in estuarine circulation. Complementing these contributions, we also focused on geometric stretching in the mouth as well as up-bay wind effects using a numerical model in idealized conditions. The geometric stretching appeared to have a limited effect on water circulation, modulating the water exchange in the mouth (see SIMU 6 vs. SIMU 2). This reduction in the mouth width accords with [5], who has suggested that constrictions block wind-induced flushing and affect along-estuary salinity gradients. The mixing mechanism in very-shallow bays due strong winds has been illustrated by [6] in a similar domain than Fangar Bay. During these episodes pycnocline is tilted and becomes nearly vertical. In these cases, mixing is fast and largely driven by shear, and the interface becomes less defined. Numerical experiments based on our simple geometric model, including density variability, are suggested for further experiments focusing on the relationship between bathymetry variability and the evolution of the hydrographic structure. Remote winds may also produce water currents which flow against the local wind within the estuary. This behavior has been observed in Delaware or Chesapeake estuaries for low frequencies and is caused by a coastal Ekman set up due to remote wind effect [4,37]. The remote wind effect seems not relevant due to the small scale and depth of the Fangar Bay, being the local frictional effects prevailing such as will be discussed in the next paragraph. Additionally, energetic wind events in Ebro delta (i.e., north-easterlies and north-westerlies winds) tends to affect a substantial percentage of the continental shelf being the wind spatial variability small [38]. However, the implementation of a nested numerical model using real configuration may solve the remote wind effect in Fangar Bay.

Observations in Fangar Bay indicated that the water circulation was complex during calm periods: current velocities were very small and lacked a clear pattern (see Figures 5, 8 and 9), consistent with other contributions mentioned above. Momentum balance can facilitate analysis of the mechanism governing water circulation [38]. In very shallow domains, the bottom frictional term ($\tau^B$) in depth-averaged momentum balance equations (i.e., $\tau^B/\rho H$) may prove relevant (i.e., $H^{-1}$ relation), indicating a substantial effect on water fluctuations. This means that local bathymetric disturbances may have a relevant role in modifying the flow. Similar findings were attained by Noble et al. [9], observing how a complex, non-linear residual force leads to a stratified estuary in the case of Mobile (USA). These studies reveal that during calm or weak wind periods, being shallow and narrow areas, estuarine circulation is weak and occurs at the expense of frictional parameters, rendering a stable circulation pattern non-existent.

Although water circulation in the inner Fangar Bay proved chaotic and complex, periods of wind intensity of up 9 m·s$^{-1}$ substantially modified water circulation within the Bay's basin. The effects of the energetic NW winds (up-bay wind) on the hydrodynamics could be clearly observed in the results. The observational results in Fangar Bay revealed that mixing occurred when the wind was equal to 10 m·s$^{-1}$ (Beaufort scale 5), as seen in Figure 10c,d. Weaver et al. [39] similarly observed that only a constant high-magnitude wind was capable of flushing the Indian River Lagoon, while Coogan et al. [35] noted that in Mobile Bay, winds of 10–15 m·s$^{-1}$ could break the strong stratification and completely mix the water column until strong river discharges re-stratified the water again. Furthermore, Geyer [5] obtained data from two estuaries on Cape Cod (USA) where weak circulation during onshore winds was altered by wind force, inhibiting circulation in the estuary. Mixing was even observed during summer, when temperatures reached 28 °C (Figure 8f). Persistent warm conditions in Fangar Bay have been associated with increases in mussel mortality and oyster growth [40]. The fact that the NW winds cool the water, as observed in the Figures 8 and 9, in a relatively short time aid

aquaculture activity. Increased warm periods in recent years represent a significant economic threat to both aquaculture producers in the bay and authorities in the area.

Wind forcing in shallow waters generates stress and introduces turbulence directly to the water column. The data analysis in Fangar Bay revealed that when the rice channels were opened during calm periods, freshwater inflow predominated and stratification was observed in the water column. Homogenization of the water column occurred during strong NW winds (up-bay wind) episodes. The same was true in periods of closed channels, when stratification was less intense. The large-scale response to wind forced may be characterized using the Wedderburn number [6,41], estimated according to the stabilization effect of the stratification and the destabilization effect of the wind:

$$W = (\Delta\rho g H^2)/L\tau_W \tag{1}$$

where $\tau_w$ = wind stress along the channel, $L$ = length of the channel, $\Delta\rho$ = density change over $L$, $H$ = channel depth and $g$ = gravitational acceleration. For a wind of 12 m·s$^{-1}$, W was equal to 0.68 (i.e., W < 1), so the wind produced a rapid, shear-driven mixture as the pycnocline tilted to become almost vertical, with the consequent development of horizontal density gradients (see Figure 10). In the case of weak winds (6 m·s$^{-1}$), the result was W equal to 2.75 (W >> 1), so in this case the pycnocline deepened slowly due to stirring. In the observations presented here, it was seen that the episodes of wind at Beaufort scale 5 came from the NW with intensities of between 10–12 m·s$^{-1}$, leading to mixing. However, being a small-scale and shallow bay, winds greater than 9 m·s$^{-1}$ may have been able to homogenize the water column. This could clearly be observed during episode E1, when winds rapidly passed from 5 to 10 m·s$^{-1}$ and were able to stimulate mixing. When there were breezes (weaker winds, ≤6 m·s$^{-1}$), the bay remained stratified. These results agreed with the calculations of the Wedderurn number, confirming that a moderately strong wind produces bay-wide mixing, due to the shallowness of the basin (4 m). For deeper depths (for example 10 m), the wind necessary to stimulate shear-driven mixing would be about 25 m·s$^{-1}$ (i.e., a wind of level 10 on the Beaufort scale). In larger bays, it is not possible for the entire water column to be mixed due to the greater role played by earth rotation and strong density gradients [3,10].

The Wedderburn number assumes that the wind is constant and blows from a longer cut-off ($T_c$). $T_c$ was the length of time needed for the effects of the boundaries to propagate to the open end of the estuary [11]:

$$T_c = \frac{L}{\sqrt{g'h}} \tag{2}$$

where L was the length of the basin, $g' = g\Delta\rho/\rho_0$ was the reduced gravity, $\Delta\rho$ represented the top-bottom density difference, $\rho_0$ was a reference water density and $h$ was the average depth. The relaxation times of the horizontal density gradients (cut-off times) for density differences of ~3 kg·m$^{-3}$, whose average depth was 2 m, were about 7 h. This means that a wind blowing during this time causes bay-wide water mixing. Therefore, in Fangar Bay, winds from the NW that usually blow for more than one day will cause mixing. The sea breeze, which is of lower intensity and blows for a shorter period than the cut-off period (Tc), is not capable of causing this mixing, as can be seen in Figures 8 and 9. After the wind ceases, the system reverts to its original state, with horizontal isopycnals. The time scale necessary to bring the system back to its original state could be calculated as [11]:

$$T_r = UL/g'H \tag{3}$$

where $H$ was the depth of the estuary and $U$ was the longitudinal velocity generated during the relaxation process, calculated in turn as:

$$U = \sqrt{g'H} \tag{4}$$

Given that *L* was small, the order of magnitude was of just a few hours. Specifically, for Fangar Bay, $T_r$ ranges of five to nine hours implied very short relaxation periods in comparison to larger domains. A visual inspection suggested a time lag between wind decay and the apparition of estuarine circulation in Fangar Bay of this order of magnitude (see Figure 8).

The water renewal has a relevant influence on water quality within the bay. A simple assessment of the residence time may be computed dividing the volume of water by the flow rate that comes out of the bay (calculated from the velocities in Figure 12). In these cases, we obtain residence times of 4 d for SIMU 2 and 4, and 1 day for SIMU 1 and 5 and respectively. The rest of the simulations are in the same order of magnitude. From the observed change flow (Figures 8 and 9) the average residence time is estimated in 4 d. The study carried out by F-Pedrera Balsells et al. [42] compute the residence times in the bay in more detail and some possible modifications to help in the bay management.

Empirical orthogonal functions (EOF) estimated for the alongshore water current directions from the entire vertical profile were used to identify the modal variability in the water current measurements (Figure 13a–d). EOFs allow multivariable data to be decomposed into its main components. Along-axis EOF analysis depicted the prevalence of the first component in both periods: 58/76% (Mouth/Bay) in FANGAR-I and 80/90% (Mouth/Bay) in FANGAR-II. During weak wind or calm periods (i.e. most of the time in FANGAR-I), the main components of the EOF analysis showed crossing-zero (suggesting stratification) with baroclinic conditions (see Figure 13a–c). However, during periods of intense winds, as was true of FANGAR-II, the first component revealed a mixing situation, seen especially at the inner bay B point, with barotropic flows.

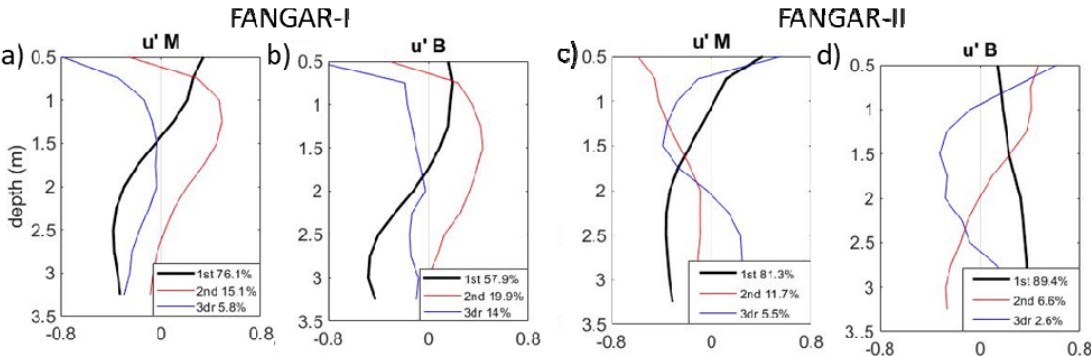

**Figure 13.** Empirical orthogonal functions (EOF) analysis for low-frequency filtered data during FANGAR-I (**a**,**b**) and FANGAR-II (**c**,**d**) along-shore currents. The legend shows the corresponding percentage of the explained variability.

Rotational effects in small bays and estuaries were expected to be small. Using the geometry of Fangar Bay, the Kelvin number was estimated, which considers the characteristic width of the estuary and the internal radius of deformation [43]:

$$K_e = B/R_{in} \qquad (5)$$

where $R_{in}$ was equal to $(g'H)^{(1/2)}/f$, $g'$ = reduced gravity, $H$ = channel depth and $f$ = Coriolis force. Rotational effects play a secondary role on the basin circulation when $K^e < 1$, and they can be considered negligible when $K^e << 1$. In the case of Fangar Bay, which is only 2 km wide and 4 m deep, $K^e = 0.39$ (<<1) suggesting that the Coriolis effect is of little importance in terms of water circulation. This was supported by a comparison of SIMU 1 and SIMU 8 (Figure 11a,h), where the differences in water currents were negligible.

Questions remain open from observational analysis, in particular the hydrodynamic response during downwind and the spatial variability of the currents. From a numerical point of view, numerical modeling effort may include the bottom roughness sensitivity test, longitudinal and vertical density

variations, sea-level oscillations, effect density gradients and baroclinic process due to river discharge sources. The numerical model implemented here is focused on solving the bathymetric effect which affect the wind-driven water circulation due to its shallowness under idealized perspective. So, realistic model implementation is a challenge to evaluate some remaining questions addressed before.

## 5. Conclusions

The dynamics in small-scale, shallow and micro-tidal estuaries were found to be characterized by very complex circulation in the inner part because of the effectiveness of local wind forcing, as well as a tendency towards stratification due to freshwater inputs that break down in episodes of intense wind lasting a few hours ($O\sim10^1$ h). Once the winds stop blowing, the system returns to its original state with $O\sim10^1$ h, so they are systems with very short relaxation periods. During these episodes of intense winds, bathymetry effects may produce an axially symmetrical transverse structure with outflow on the axis of the central channel, opposite to the wind direction, alongside inflow in shallow lateral areas. Furthermore, being a small-scale bay (2 km wide) and very shallow (average depth 2 m) these winds are capable of mixing all the water inside the bay, and being a narrow domain, the rotation effect is negligible. According to the data obtained from field campaigns and numerical idealized experiments in Fangar Bay, we suggest that a small-scale and shallow bay can be defined as being of 20 km$^2$ area and about 4 m depth, respectively.

**Author Contributions:** Conceptualization, methodology, M.F-P.B., M.G. and M.E.; software, M.F-P.B., M.G. and M.E.; validation, M.G. and M.E.; formal analysis, M.F-P.B.; investigation, M.F-P.B., P.C., M.G. and M.E.; resources, M.G. and M.E.; data curation, M.F-P.B.; writing—original draft preparation, M.F-P.B.; writing—review and editing, M.G., M.E. and A.S.-A.; visualization, M.G.; supervision, M.G., M.E. and A.S.-A.; project administration, M.E. and A.S.-A.; funding acquisition, M.E. and A.S.-A. All authors have read and agreed to the published version of the manuscript.

**Funding:** This research received funding from the Ecosistema-BC CTM2017-84275-R AEI/FEDER, UE.

**Acknowledgments:** We would like to thank Jordi Cateura and Joaquim Sospedra (LIM-UPC, Barcelona, Spain) and Margarita Fernandez (IRTA, Sant Carles de la Ràpita, Spain) for the data acquisition campaign and Alfredo Aretxabaleta (USGS, Woods Hole, USA) for his suggestion in previous versions of the manuscript. Also by the Government of Catalonia, in the framework of the FEMP-FANGAR2017 project and by the Direcció General de Pesca of the Government of Catalonia (DGP). Data from that project used in the current paper are archived in a public repository accessible at https://doi.org/10.5281/zenodo.3756624. As a group, we would like to thank the Secretaria d'Universitats i Recerca del Departament d'Economia i Coneixement de la Generalitat de Catalunya (2017SGR773).

**Conflicts of Interest:** The authors declare no conflict of interest. The funders had no role in the design of the study; in the collection, analyses, or interpretation of data; in the writing of the manuscript, or in the decision to publish the results.

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
