# Peer review of "Wind-Driven Hydrodynamics in the Shallow, Micro-Tidal Estuary at the Fangar Bay (Ebro Delta, NW Mediterranean Sea)"

_applsci, doi:10.3390/app10196952_

Round 1
Reviewer 1 Report
The paper has been an interesting read about the hydrodynamics of the Fangar Bay. It analyses observation data from two field campaigns to show the circulation in during calm conditions and stormy NW wind conditions. The observational description and analysis is very well done.
The numerical component aims to show the effect of bathymetry and constrictions on the circulation (as function of wind directions). However, the modelling component is not given the same level of details or analysis as the observations. If the modelling was only looking at the wind effect, there should be a major consideration (or why not) of the remote wind effect (coastal water level setup by wind) and local estuary wind setup (estuary water level setup by wind) and stratification. See Garvine 1985 in JGR. He suggests that in observations where the estuary current direction is opposite to the prevailing wind field is due to the remote wind effect dominating the local effect. It is hard to see if this applies to the model because the timescale of the model scenarios aren't discussed here. In addition, the model results are not explained or shown sufficiently, i.e., why the bathymetric changes imposed that circulation. This could be done using the momentum terms (advection, frictional, acceleration etc) output by ROMS.
Finer Details:
Lines 132-138: The domain grid is said to be 37x27 with a dx=70m, so it should be 2.5km x 1.9km, however the text says 5km x 2km.
Lines 132-138: The boundary effects are not mentioned. If Figure 2 is the full domain, then the offshore boundaries are too close to the study domain and boundary noise can propagate into the estuary.
Lines 132-138: While inside the estuary, the Coriolis effect is small, it can be large outside in the coastal domain. So if the boundary is too close this effect is constrained by the boundaries.
Lines 132-138: During the numerical experiment analysis, are the surface and bottom layers simply the 1st and 10th layers, or integration of certain layers. It should be made early in this section.
Lines 132-138: Maybe include a plot of the cross-section of the vertical sigma layers, so readers have a idea of the thickness of the layers.
Line 146: Is the model applied as a barotropic model? Why? The observations show that vertical stratification is important in Fangar Bay.
Lines 151: Why is this value chosen? How sensitive are the model results to this?
Line 151: What is the frictional coefficient set and how sensitive is the model results to that?
Line 151: How long the each simulation? Spin up before winds applied or not? Duration of the wind? Does the wind slowly ramp up to the constant value or is the wind is initially ramped up as a dirac function to the constant value? How long after the wind is applied, do the model results get extracted to avoid transient noise.
Line 151: Up-estuary and down-estuary winds can cause alongshore coastal flows. SIMU 1 & 8 (off) don't seem to show this effect. Does this mean that the model are extracted before an inertial period has passed after the wind is applied?
Line 309 seems to say the opposite of the results>>
"up-bay flow occurs in the channel and down-bay flow in the lateral shoals for up-estuary wind pulses", did it mean for down-estuary winds? SIMU 2 is up-est winds, which shows up-bay flows in shoals and down-bay in channel.
Line 270-272: SIMU 4 seems to be missing here.
Line 308-320: There is no discussion of why the model results are what they are when bathymetric effects are considered. This is where Garvine 1985 (remote & local effect of winds) would be a useful guide.
Reviewer 2 Report
RE: Hydrodynamics in micro-tidal and small-scale estuaries: The Fangar case (Ebro Delta, NW Mediterranean sea).
Dear Authors,
This article integrates seasonal oceanographic deployments and intensive field campaigns, along with simplified numerical modelling to characterize the main circulation patterns in a small, shallow and micro-tidal estuary. The study case is Fangar at the Ebro Delta.
I think the study is relevant and it fits the scopes of the journal. However, I have a mixed impression on the overall scope of the work. On one hand, I feel authors have the opportunity to use the in-situ observations and numerical model to go deep on the processes they identify as relevant, for instance, the strong temporal and spatial variability that systems like Fangar might experience. While on the other hand, I feel that the overall use of the observation and simplified modelling is not exploited as it could be done, making the analysis rather too limited instead of simple and robust.
My issues start with the title. The authors mention “estuaries”, whereas the only case they examine in the Fangar bay. So I would stick to "estuary" unless they wish to develop a general study, based on modelling and theory, to understand the hydrodynamics in micro-tidal, small-scale estuaries. Then, there is a parenthesis with “Ebro Delta, NW Mediterranean Sea”. This is a well-known region, so I would not make the title longer. I would use the following words
“Wind-driven hydrodynamics in the shallow, micro-tidal estuary at the Fangar bay, Ebro Delta, Catalonia”
My general comment is that authors should frame objectives, scopes, and outcomes of the manuscript early in the introduction. To be transparent, I was excepting a study of the hydrodynamics in micro-tidal and small-scale estuaries. I was thinking on the circulation process, transport, flushing processes and renewal timescales of water in the Fangar bay, decoupling the forcing mechanisms, such as wind, in-flow, micro-tides, Coriolis induced circulation (even for low Kelvin numbers), and robust description of baroclinic motions. Little of these were found, but the authors have the data and the model to explore such relevant aspects (and more) that characterise “hydrodynamics of an estuary”, and therefore to achieve the goal you are looking for.
I would like to see quantifiable results. For instance, a relevant scale to examine in the bay is the residence time and flushing process. For this, they have to compute the horizontal transport and the horizontal mass exchange across the bay. This is a meaningful quantity that could be of interest for the locals to have a measure of the water ventilation in the Fangar bay, and it could be easily obtained from either the numerical experiments or, with some hypotheses, from the ADCP data.
Section 2.2: Field campaigns.
- In general, it is useful to provide full information of the gears deployed (series numbers, setting, etc) and whose observations were used for the study. This information could be included in a table.
Section 2.3: Numerical modelling
- Why did you choose to use vertical stretched coordinates? Which type? Did you try a uniform vertical grid? Did you compare the results in terms of mixing? More information about your choices regarding the modelling setup will be welcome.
- What type of boundary conditions have you imposed in the 3D variables?
- How did you choose the bathymetry for cases Geo 2 and Geo 4, and the width of the mouth for case 3?
Section 3.1. Results: Meteorological description of Fagar Bay.
- Could you please provide time-averaged diurnal profiles of wind along the main axes of the Bay?
- Do you have information on solar radiation?
Section 3.2. Results: Hydrodynamic description of Fangar Bay.
- Do you have information about surface water elevation or tide?
- Do you have information about the river outflow at the Ebro Delta?
- L201: I do not think this sentence helps: “The data presented so far were obtained from the ADCPs without any treatment”. If you consider that the ADCP signal might contain noise or contaminated data, this sentence suggests a careless treatment of the data.
- L201-202: “The aim of this paper …” I do not like this sentence either. The goal of the paper should be up in the introduction. Of course, you have to treat and filter the data to obtain what you are looking for. I understand that this is not a technical paper that discusses a method of how to do filtering, so as a reader I would expect that the data provided on the figures were already treated. Just a brief description of the methodology used would be needed.
Section 3.3. Results: Numerical experiments.
- Do you include an inflow of freshwater? If not, please stress it. Again, what type of boundary conditions have you imposed?
- Where did you describe SIMU 4?
- Does SIMU 8 show a residual flow?
Discussion:
- The authors have not analysed the role of the micro-tidal signal. How negligible is it on the circulation of shallow estuaries as the Fargar bay?
I would suggest the authors read the following papers. They might help to give a more detailed description of the hydrodynamics of this coastal system. I would put a particular emphasis on the exchange properties in the estuary. For instance, authors could examine the exchange flow and the net transport in the estuary based on the alongshore flow measured and the numerical experiments as done in these works.
References:
- Rockwell Geyer and Parker MacCready (2014). The Estuarine Circulation, Annual Review of Fluid Mechanics 46:1, 175-19
- Ulloa, H. N., K. A. Davis, S. G. Monismith, and G. Pawlak (2018). Temporal Variability in Thermally Driven Cross-Shore Exchange: The Role of Semidiurnal Tides. Phys. Oceanogr., 48, 1513–1531, https://doi.org/10.1175/JPO-D-17-0257.1.
- Rodriguez, A. R., Giddings, S. N., & Kumar, N. (2018). Impacts of nearshore wave‐current interaction on transport and mixing of small‐scale buoyant plumes. Geophysical Research Letters, 45, 8379– 8389. https://doi.org/10.1029/2018GL078328
- Giddings, Sarah N. and Monismith, Stephen G. and Fong, Derek A. and Stacey, Mark T. (2014). Using Depth-Normalized Coordinates to Examine Mass Transport Residual Circulation in Estuaries with Large Tidal Amplitude Relative to the Mean Depth. Journal of Physical Oceanography. 44:128 - 148. : American Meteorological Society 1175/JPO-D-12-0201.1
Minor comments:
- L43 delete “some” from the sentence.
- L47-48: The last sentence of the paragraph does not emphasise that this study is aiming at explain at all. I think it does not contribute much framing the research gap this study is filling up.
- L65-67: Is this the most important outcome from the manuscript?
- L101: What type of ADCPs were used, brand, model, frequency?
- L104: What type of temperature logger were used?
- L124: Did you use ROMS only to analyse the hydrodynamic wind response in Fangar Bay?
- L264-265: “when the bathymetry variability was … “ -> ” when a varying bathymetry was … “
- L266: What do you mean by “the friction of the bottom is imposed, a very significant factor in shallow environments”.
- L291-293: “Shallow depths accentuate … “ Could you explain more in deep what kind of complex water circulation you are referring to?
- L416: I think it is correct what comes after the sentence “Rotational effects will be negligible when B is less than 5 km wide”, but the last sentence might mislead readers. Therefore, I suggest expression the effect of Coriolis in terms of the non-dimensional number Kelvin. For instance, “Rotational effects play a secondary role on the basin circulation when Ke<1, and they can be considered negligible when Ke<<1.
- L424-426: I think this sentence adds no value to the paper. It only makes us wait for future work that no one knows when it is going to be available. Additionally, this sentence defines the scientific scope of the work. If the objective is to examine the role of the wind only, you have to justify why the wind is the primary driving mechanism to investigate in the Fangar case, and perhaps it should be even included in the title.
Round 2
Reviewer 1 Report
I'm happy with the changes.